# An Enhanced Synaptic Plasticity of Electrolyte-Gated Transistors through the Tungsten Doping of an Oxide Semiconductor

Dongyu Xie [1], Xiaoci Liang [1,*], Di Geng [2], Qian Wu [3] and Chuan Liu [1]

1    The State Key Laboratory of Optoelectronic Materials and Technologies, Guangdong Province Key Laboratory of Display Material and Technology, School of Electronics and Information Technology, Sun Yat-sen University, Guangzhou 510275, China; xiedy3@mail2.sysu.edu.cn (D.X.); liuchuan5@mail.sysu.edu.cn (C.L.)
2    State Key Laboratory of Microelectronic Devices and Integrated Technology, Institute of Microelectronics, Chinese Academy of Sciences, Beijing 100045, China; digeng@ime.ac.cn
3    School of Computer and Information Engineering, Guangdong Polytechnic of Industry and Commerce, Guangzhou 510091, China; wuqian1427@gdgm.edu.cn
*    Correspondence: liangxc5@mail.sysu.edu.cn

**Abstract:** Oxide electrolyte-gated transistors have shown the ability to emulate various synaptic functions, but they still require a high gate voltage to form long-term plasticity. Here, we studied electrolyte-gated transistors based on $InO_x$ with tungsten doping ($W-InO_x$). When the tungsten-to-indium ratio increased from 0% to 7.6%, the memory window of the transfer curve increased from 0.2 V to 2 V over a small sweep range of $-2$ V to 2.5 V. Under 50 pulses with a duty cycle of 2%, the conductance of the transistor increased from 40-fold to 30,000-fold. Furthermore, the $W-InO_x$ transistor exhibited improved paired pulse facilitation and successfully passed the Pavlovian test after training. The formation of $WO_3$ within $InO_x$ and its ion intercalation into the channel may account for the enhanced synaptic plasticity.

**Keywords:** electrolyte-gated transistors; long-term potentiation; paired pulse facilitation; synaptic functions

## 1. Introduction

The electrolyte-gated transistor has been a promising candidate for neuromorphic computing due to its ability to simulate various synaptic behaviors, including short-term plasticity and long-term plasticity [1,2]. Under the stimulation of the gate bias, the ions can be adsorbed at the interface between the electrolyte and semiconductor causing the variation in channel conductance. With various relaxation times in electrostatic adsorption and electrochemical adsorption, the change in conductance decays after stimulation occurs in a short time and long time, respectively [3].

Amorphous oxide semiconductors, such as $InO_x$, IZO, and IGZO, due to their high mobility and compatibility with large-area semiconductor device processes, have been employed to fabricate electrolyte-gated transistors [4–7]. These devices have shown great performance in their application to neuromorphic computing and artificial vision. However, for achieving the transition from short-term plasticity to long-term plasticity, it is usually necessary to increase the gate voltage, which likely results from the high migration and adsorption energy barriers of ions in these oxide semiconductors. Introduction materials with low migration and adsorption energy barriers may provide a solution to this problem. Tungsten trioxide ($WO_3$), due to its active d-orbital and layered structure, prefers intercalation by hydrogen ions, lithium ion, and sodium ion at $[WO_6]$-octahedral sites under potential bias. Among these ions, the diffusion barrier for hydrogen ions in $WO_3$ is relatively low because of the smallest size [8–11]. This indicates that $WO_3$ is likely suitable as a dopant to improve the plasticity performance.

In this work, we studied the effect of tungsten doping in $InO_x$ electrolyte-gated transistors. The $InO_x$ transistor doped with tungsten showed significant enhancements

in long-term plasticity compared with the InO$_x$ transistor, characterized by an expansive memory window from 0.2 V to 2 V and increased conductance from 40-fold to 30,000-fold. Synaptic functions, including the paired pulse facilitation and Pavlovian conditioning, were also enhanced by tungsten doping.

## 2. Materials and Methods

The 0.2 mol/L AlO$_x$ precursor solution was synthesized by dissolving aluminum nitrate hydrate (Aladdin, Shanghai, China), nitric acid (Alfa Aesar, Ward Hill, MA, USA), and ammonium hydroxide (50% *v/v*, Alfa Aesar, Ward Hill, MA, USA) in water. The 0.15 mol/L InO$_x$ precursor solution was synthesized by dissolving indium nitrate hydrate (Aladdin, Shanghai, China) in 2-Methoxyethanol (Sigma-Aldrich, St. Louis, MO, USA). The 0.1 mol/L WCl$_6$ solution was synthesized by dissolving tungsten hexachloride (Aladdin, Shanghai, China) in 2-methoxyethanol. The InOx with tungsten doping (W-InO$_x$) precursor solution was synthesized by mixing the InO$_x$ precursor solution and the WCl$_6$ solution with volume ratios of 40:3 and 20:3, respectively. For the fabrication of the transistors, the AlO$_x$ precursor solution was spin-coated on a heavily doped silicon wafer and annealed on a hot plate at 300 °C for 30 min; the spin-coating process was repeated five times. The InO$_x$ and W-InO$_x$ precursor solutions were spin-coated, respectively, on the AlO$_x$ and annealed on a hot plate at 280 °C for 1.5 h. After the photolithography process and etching with hydrochloric acid, the channel layer was patterned. Finally, Al source/drain electrodes were deposited by thermal evaporation to fabricate the bottom-gate, top-contact transistors. The schematic illustration of the transistor is shown in Figure 1a.

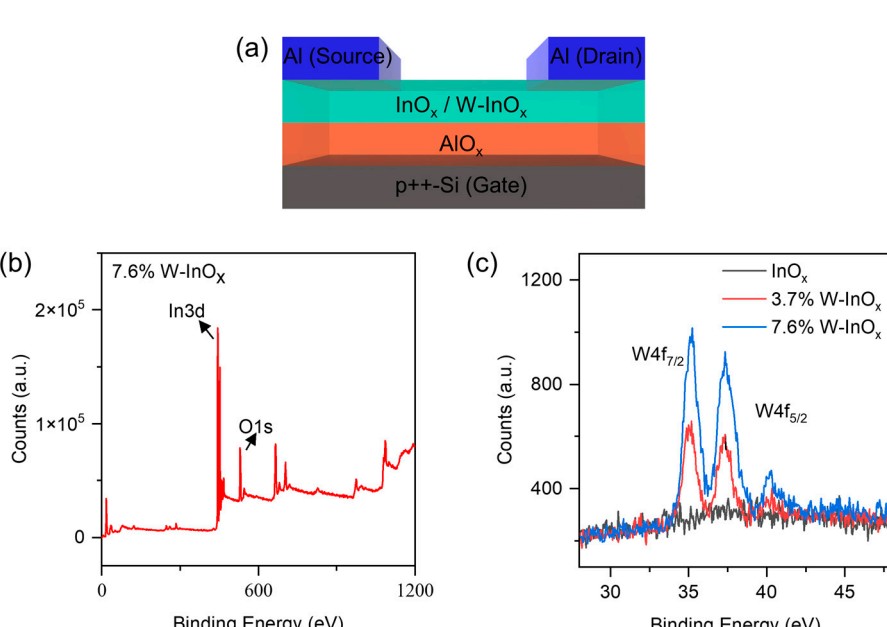

**Figure 1.** (**a**) Scheme of the electrolyte-gated transistor using the AlO$_x$ as an electrolyte layer and the W-InO$_x$ as a semiconductor layer. (**b**) XPS survey for the 7.6% W-InO$_x$ film. (**c**) XPS W4f core-level for the InO$_x$ films with various tungsten concentrations.

## 3. Results

In a previous study, the solution-processed AlO$_x$ films were confirmed to act as an electrolyte with mobile hydrogen ions [12]. Due to the formation of the electric double layer (EDL) and ionic electrochemical adsorption, the conductance of the channel can be adjusted, which leads to the synaptic plasticity. To promote the synaptic plasticity, we fabricated the InO$_x$ films with tungsten doping (W-InO$_x$) as a channel layer. The chemical composition of these films was determined by X-ray photoelectron spectroscopy (XPS). The atom ratios of W to In were 3.7:100 and 7.6:100 in the W-InO$_x$ films with various tungsten concentrations, respectively. Figure 1b shows the survey spectrum of the 7.6% W-InO$_x$

film. The peaks at about 445 eV for In3d and about 530 eV for O1s suggest that the films consist of the elements In and O. As shown in Figure 1c, the W4f spectrum exhibits two peaks at 35.3 eV and 37.4 eV in both the 3.7% and the 7.6% W-InO$_x$, which correspond to the W4f$_{7/2}$ and W4f$_{5/2}$ peaks of WO$_3$, respectively [13,14]. The result indicates the valence state of W is +6. For the InO$_x$ film, the absence of the W4f peaks indicates that there are no W elements in the film.

Figure 2a shows the transfer curves with various tungsten concentrations at $V_D$ = 1 V. The anticlockwise hysteresis of the transfer curve increases with increasing tungsten concentration. The hysteresis windows $\Delta V_G$ at $I_D$ from 10 nA to 50 nA are extracted as the memory windows. As shown in Figure 2b, the memory windows (MWs) from $I_D$ = 10 nA to 50 nA for the InO$_x$, 3.7% W-InO$_x$, and 7.6% W-InO$_x$ transistors are 0.215 V, 0.579 V, and 1.998 V, respectively, across the sweep range of −2 V to 2.5 V. The 7.6% W-InO$_x$ transistor shows a wide memory window, accounting for 44% of the sweep range. Figure 2c,d shows the responses to consecutive pulses in W-InO$_x$ transistors. The normalized excitatory postsynaptic currents (EPSCs) were calculated as $(I_D - I_{min})/(I_{max} - I_{min})$. The decay current rises from 0.016 to 0.166 as the W to In ratio increases from 0% to 7.6% after applying positive gate pulses for 30 s. The decay current declines from −0.089 to −0.304 after applying negative gate pulses for 30 s. The enhanced amplitude of variation of the decay current in the 7.6% W-InO$_x$ transistor, following both positive and negative pulses, signifies a transition from short-term to long-term plasticity. For the 7.6% W-InO$_x$ transistor, under continuous gate pulse stimulation, the $I_D$ gradually increases and exhibits short-term plasticity. After the pulses, the increased $I_D$ can be maintained for a long time, exhibiting long-term plasticity. For the InO$_x$ transistor, after the same gate pulse stimulation, the $I_D$ decays in a short time and only exhibits short-term plasticity.

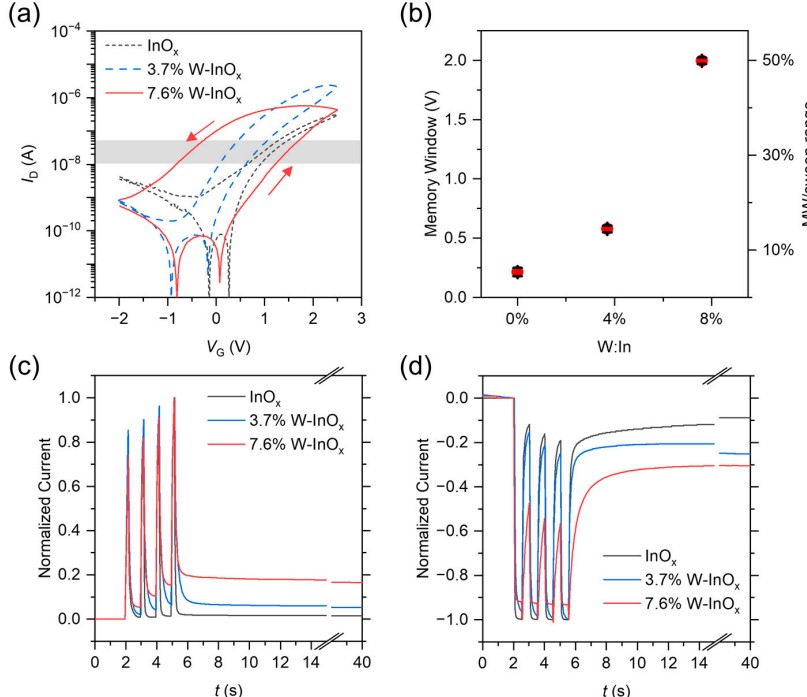

**Figure 2.** (**a**) The transfer curves of the W-InO$_x$ transistor with various concentrations of tungsten. The gray band at $I_D$ from 10 nA to 50 nA is the area for extracting the memory window. The arrows mark the scanning direction of curve. (**b**) The memory window (MW) extracted from the $\Delta V_G$ in forward and backward curves at $I_D$ from 10 nA to 50 nA. The value (red dots) and error bar (black sticks) are the average and the standard deviation of $\Delta V_G$, respectively. The long-term potentiation (**c**) and depression (**d**) of the W-InO$_x$ transistors under continuous pulses. The pulse amplitude, width, and period in (**c**) are 4 V, 200 ms, and 1 s, respectively, and are −4 V, 800 ms, and 1 s, respectively, in (**d**).

Paired pulse facilitation (PPF) is a neuron behavior consisting of two EPSCs elicited by a pair of pulses, where the latter EPSC is larger than the former one [15]. As shown in Figure 3a, two pulses with an interval time $\Delta t$ of 0.3 s were applied to the gate electrode. The PPF index was calculated as the PPF Index = $I_2/I_1$. Figure 3b illustrates that the PPF index increases as the $\Delta t$ decreases, emulating the synaptic short-term memory [16]. With an increase in the W-to-In ratio from 0% to 7.6%, the PPF index at $\Delta t$ = 50 ms rises from 1.17 to 2.14, suggesting an enhancement of the short-term memory. The PPF curve was fitted with a double-phase exponential function defined as follows: PPF Index = $1 + C_1 \exp(-\Delta t/\tau_1) + C_2 \exp(-\Delta t/\tau_2)$ [17]. $\tau_1$ and $\tau_2$ are the characteristic relaxation times for fast ionic decay and slow ionic decay, respectively. $\tau_1$ corresponds to the short relaxation time of residual ions after the first stimulation. $\tau_2$ corresponds to the long relaxation time of ions after the second stimulation [18]. For the InO$_x$ transistor, $\tau_1$ and $\tau_2$ are estimated to be 63.04 ms and 869.47 ms, respectively. For the 7.6% W-InO$_x$ transistor, the corresponding values are 407.43 ms and 2168.18 ms, respectively. The increased $\tau_1$ and $\tau_2$ values for the 7.6% W-InO$_x$ transistors indicate a reduced forgetting rate in the short-term memory [19].

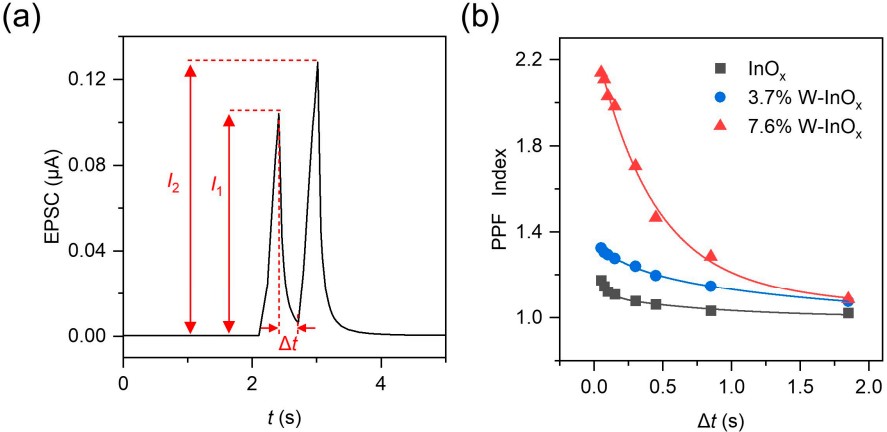

**Figure 3.** (**a**) EPSCs at $V_D$ = 1 V triggered by a pair of gate voltage pulses; $I_1$ and $I_2$ are the current values at the end of the first and the second pulses. The pulse amplitude, width, and period are 2 V, 0.3 s, and 0.6 s, respectively. (**b**) PPF index as a function of the interval time $\Delta t$ and fitted by a double-phase exponential function.

The enhancement of long-term plasticity and the PPF behavior in the 7.6% W-InO$_x$ transistor is likely due to the formation of the WO$_3$ and hydrogen ion intercalation within WO$_3$. As shown in Figure 4, without W doping, the electrochemical accumulation of ions driven by the high gate bias decays quickly after the gate bias is removed, resulting in low long-term potentiation and a short relaxation time in PPF. With W doping, the electrochemical absorption and the proton intercalation in WO$_3$ can synergistically improve the storage of the hydrogen ions in the channel layer. The hydrogen ions can act as the donor in the InO$_x$ and WO$_3$ and increase the channel conductance [11,20]. The more adsorbed hydrogen ions result in higher long-term potentiation and an extended relaxation time in PPF [7,10,21].

The longer relaxation time can lead to a longer memory duration and higher amplitudes of the potentiation and depression. As shown in Figure 5a–c, as the W-to-In ratio increases from 0% to 7.6%, the conductance of the transistor increases from 40-fold to 30,000-fold after applying 50 positive gate pulses with a 2% duty ratio. Furthermore, as the W-to-In ratio increases, the conductance of the transistor decreases from 0.91 to 0.39 compared to the initial value after applying 50 negative gate pulses. The enhancement in the potentiation and depression behaviors for the 7.6% W-InO$_x$ transistor, stimulated by such sparse pulses, suggests that tungsten doping can effectively prolong the memory time in the channel.

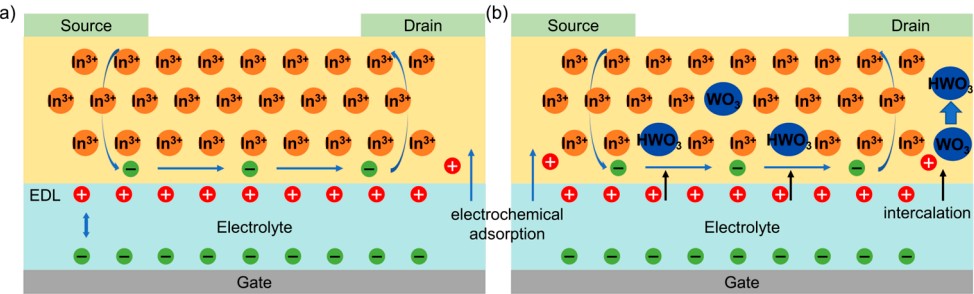

**Figure 4.** (**a**) Schematic diagram of the EDL and electrochemical adsorption in the InO$_x$ electrolyte-gated transistor. (**b**) Schematic diagram of the electrochemical adsorption and ion intercalation in the W-InO$_x$ electrolyte-gated transistor.

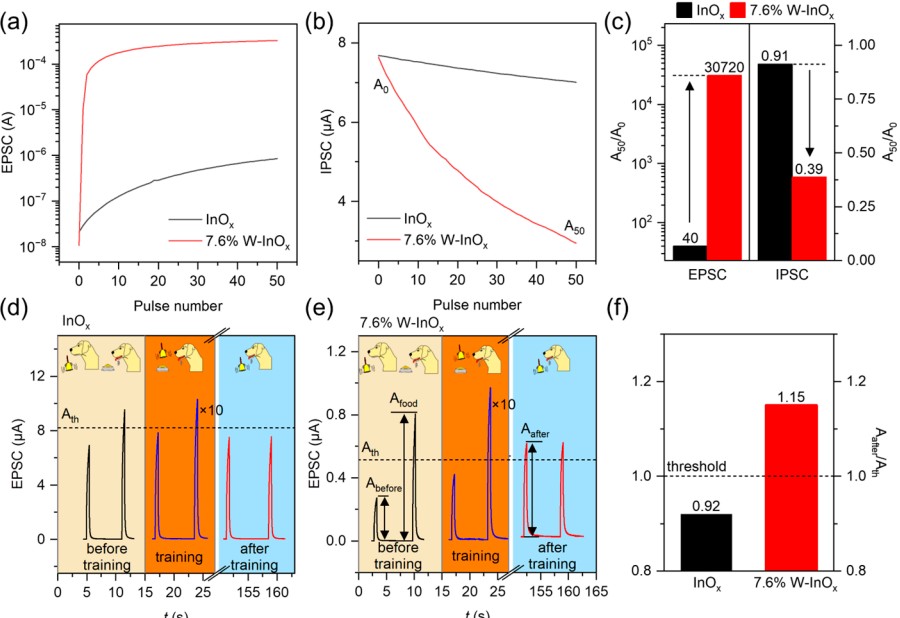

**Figure 5.** The excitatory postsynaptic current (**a**) and the inhibitory postsynaptic current (**b**) as functions of the pulse number with a pulse period of 10 s. The pulse amplitude and width are 4.5 V and 0.2 s for (**a**) and −4.5 V and 0.8 s for (**b**), respectively. (**c**) The ratio of the current after the last pulse (A$_{50}$) to the current before the first pulse (A$_0$). Pavlovian experiment for the InO$_x$ transistor (**d**) and the 7.6% W-InO$_x$ transistor (**e**). The pulse amplitudes correspond to 2 V for the bell signal and 3 V for the food signal, with a pulse width of 0.5 s. (**f**) The ratio of the response to the bell signal after training (A$_{after}$) to the threshold (A$_{th}$).

To demonstrate the connection between learning and memory, the Pavlovian experiment was performed [22,23]. Gate pulses with amplitudes of 2 V and 3 V were utilized to simulate the bell ring and the food signals, respectively. As shown in Figure 5d–f, before training, the bell signal caused a low EPSC (A$_{before}$), while the food signal caused a high EPSC (A$_{food}$). We set the average of A$_{before}$ and A$_{food}$ as the threshold (A$_{th}$). When the EPSC stimulated by the bell signal exceeds the threshold, it means that the dog salivated. During the training, the food signal was delayed by 7 s compared to the bell signal. Within a period of 15 s, both the food signal and the bell signal were used for training once. After 10 trainings with the bell ring and the food, the EPSC stimulated only by the bell signal (A$_{after}$) clearly increased to 1.15 times of A$_{th}$ for the 7.6% W-InO$_x$ transistor, whereas for the InO$_x$ transistor, the EPSC did not reach the threshold. The successful training indicates that the W-InO$_x$ transistor has better learning and memory capabilities.

## 4. Conclusions

The synaptic performance of the $InO_x$ electrolyte-gated transistors with various W concentrations was studied. As the W-to-In ratio increased from 0% to 7.6%, the transistor showed an increasing hysteresis window from 0.2 V to 2 V with a sweep range of $-2$ V to 2.5 V in the transfer curve, and the conductance increased from 40-fold to 30,000-fold after applying 50 positive pulses with a 2% duty ratio. With tungsten doping, the synaptic performance was improved with a higher PPF index and longer relaxation time in PPF, and a successful training in the Pavlovian experiment was performed compared with those without tungsten doping. The formation of $WO_3$ and its resulting ion intercalation may be the cause of this enhancement.

**Author Contributions:** Device design, D.X.; fabrication, D.X.; characterization, D.X.; writing—original draft preparation, D.X., X.L., D.G., Q.W., C.L. All authors have read and agreed to the published version of the manuscript.

**Funding:** This research was funded by the National Key Research and Development Program of China (2022YFB3603901) and the China National Postdoctoral Program for Innovative Talents (BX20230439).

**Data Availability Statement:** The data presented in this study are available from the corresponding author upon reasonable request.

**Conflicts of Interest:** The authors declare no conflicts of interest.

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
