# Peer review of "An Enhanced Synaptic Plasticity of Electrolyte-Gated Transistors through the Tungsten Doping of an Oxide Semiconductor"

_electronics, doi:10.3390/electronics13081485_

Round 1

Reviewer 1 Report

Comments and Suggestions for Authors

In this work, the authors improved the memory effect in electrolyte-gated transistors by increasing the tungstene doping in InOx films. The results are sound and are well explained. Then, I recommend publication after minor revision.

Please, add in the manuscript a broad explanation of the following issues:

1.  Line 108. How is the next sentence justified?: "signifies a transition from short-term to long-term plasticity."

2. Line 163. Be more specific about how the Pavlovian experiment was performed. How did you apply two gate voltage pulses simultaneously?

Reviewer 2 Report

Comments and Suggestions for Authors

This paper presented a very interesting study regarding demonstration of synaptic plasticity by three terminal transistor device with W-doped InOx channel layer. The authors attributed the mechanism coming from protons which were produced in WOx.  However, there are several insufficient explanations. The author are required to revise the manuscript regarding the following points:

1)In Fig.2 (b), memory window is shown.  However, a definition of memory window is not clear. It is important to describe how the authors determined the memory window more in detail. Is it possible to show what the memory window is schematically using Fig. 2(a) ?

 2)PPF is introduced in Line 109.  It is better to write down full spellings such that PPF (paired pulse facilitation).

 3)Line 116:  PPF index shown in Fig.3 (b) has 2 components of time dependences.  What kind of phenomena are the origins of the two relaxation times t1 and t2 ?

 4)Fig.4 (b) is almost the same as Fig. 4 (a) except for the only one tungsten ion.  It is mandatory to revise Fig. 4 (b). In the line 133-134, It is written that proton intercalation in WO3 can improve storage of hydrogen ions in the channel layer. This hypothesis seems to be probable. How about to draw schematics of this concept?

Comments on the Quality of English Language

In line 182-183;  successful training in Pavlovian experiment compared with those without tungsten doping. 

It is better to add “was performed” in the last part of the sentence.
